# Glutamine Deficiency Promotes Immune and Endothelial Cell Dysfunction in COVID-19

**DOI:** 10.3390/ijms24087593

**Published:** 2023-04-20

**Authors:** William Durante

**Affiliations:** Department of Medical Pharmacology and Physiology, University of Missouri, Columbia, MO 65212, USA; durantew@health.missouri.edu

**Keywords:** COVID-19, glutamine, glutaminase, ammonia, heme oxygenase-1, immune and endothelial dysfunction, coagulopathy, vascular disease

## Abstract

The coronavirus disease 2019 (COVID-19) pandemic has caused the death of almost 7 million people worldwide. While vaccinations and new antiviral drugs have greatly reduced the number of COVID-19 cases, there remains a need for additional therapeutic strategies to combat this deadly disease. Accumulating clinical data have discovered a deficiency of circulating glutamine in patients with COVID-19 that associates with disease severity. Glutamine is a semi-essential amino acid that is metabolized to a plethora of metabolites that serve as central modulators of immune and endothelial cell function. A majority of glutamine is metabolized to glutamate and ammonia by the mitochondrial enzyme glutaminase (GLS). Notably, GLS activity is upregulated in COVID-19, favoring the catabolism of glutamine. This disturbance in glutamine metabolism may provoke immune and endothelial cell dysfunction that contributes to the development of severe infection, inflammation, oxidative stress, vasospasm, and coagulopathy, which leads to vascular occlusion, multi-organ failure, and death. Strategies that restore the plasma concentration of glutamine, its metabolites, and/or its downstream effectors, in conjunction with antiviral drugs, represent a promising therapeutic approach that may restore immune and endothelial cell function and prevent the development of occlusive vascular disease in patients stricken with COVID-19.

## 1. Introduction

Severe acute respiratory syndrome coronavirus 2 (SARS-CoV-2) is the etiological factor responsible for the coronavirus disease 2019 (COVID-19) pandemic. This virus is a global public health concern that has infected more than 760 million people and claimed over 6.8 million lives worldwide, as of March 2023 [1]. SARS-CoV-2 is an enveloped virus containing a positive-strand RNA genome that is transmitted by the inhalation of respiratory droplets [2]. The coronavirus enters the host through airway and alveolar epithelial cells in the lung that express the membrane receptor angiotensin-converting enzyme 2 (ACE2) [3]. The virus infects host cells via its surface spike protein that binds to ACE2 through its receptor-binding domain, where it is proteolytically activated by host proteases, allowing the fusion of the viral and host membranes. SARS-CoV-2 is then internalized by endocytosis, and viral RNA is released for replication and translation by the host cell machinery. Following assembly, new viral particles are released by dying infected cells. In most cases, the destruction of lung cells by the virus initiates a local immune response that evokes the generation of anti-viral cytokines and primes adaptive T and B cell immune responses that resolve the infection resulting in mild flu-like symptoms such as fever, cough, and fatigue [4,5,6]. However, in certain instances, a dysfunctional immune response arises where a proinflammatory feedforward loop is created, triggering a cytokine storm that promotes extensive pulmonary inflammation, leading to severe pneumonia and adult respiratory distress syndrome (ARDS). Furthermore, a leak of inflammatory cytokines into the circulation induces endothelial cell (EC) and smooth muscle cell (SMC) dysfunction, oxidative stress, and coagulopathy, which contributes to multiorgan failure and high rates of mortality in COVID-19 patients. Although vaccination programs and recently approved antivirals have drastically curtailed SARS-CoV-2 infection and disease severity, mutant viruses are emerging under the pressure of host immunity that instigate new waves of infection, highlighting the need for additional therapeutic strategies to fight this deadly pandemic.

Recent reviews have described the possible efficacy of nutraceutical approaches in reducing disease severity in patients with COVID-19, based on their ability to improve immune function, malnutrition, and vascular disease in stricken individuals [7,8,9,10]. Among nutrients, glutamine is a highly versatile molecule that is of vital importance in intermediary metabolism, interorgan nitrogen exchange, ammonia detoxification, and pH homeostasis [11]. In addition, glutamine is a major regulator of immune function, influencing both myeloid and lymphoid cells [12,13,14,15]. Clinical studies indicate that glutamine supplementation improves the immune system in patients with burn injury, gastrointestinal surgical operations, and in critically ill patients [16,17]. This is reflected by increases in lymphocyte number and function and reductions in infectious complications, hospital stay, morbidity, and mortality. Favorable clinical outcomes with glutamine have also been reported in patients undergoing major abdominal surgery or receiving bone marrow transplants [18,19,20]. Moreover, emerging work reveals that glutamine plays a fundamental role in preserving vascular health by modulating—EC and SMC function [21,22]. Thus, a reduced availability of glutamine may impair immune function and increase the susceptibility of humans to infections and vascular disease. Intriguingly, metabolic analyses have identified low levels of glutamine in COVID-19 patients, suggesting that the rewiring of glutamine metabolism by SARS-CoV-2 may contribute to poor outcomes in these patients [23]. In this review, we characterize alterations in circulating glutamine in COVID-19 patients and discuss how the depletion of glutamine may contribute to immune and vascular cell dysfunction and its associated risk of lethal pulmonary and vascular complications. Furthermore, it investigates putative therapeutic strategies that target glutamine in COVID-19.

## 2. Glutamine Metabolism

Glutamine is a neutral, conditionally essential amino acid that is indispensable during periods of rapid growth or in pathological states such as trauma, sepsis, and infection [12,13,14,15]. It is the most abundant amino acid in the blood and is used as a nitrogen and carbon source to synthesize amino acids, lipids, and nucleic acids. Glutamine enters cells via multiple membrane transporters that make up the solute carrier (SLC) superfamily (Figure 1). At least fourteen glutamine transporters have been identified at the molecular level that belong to four distinct families: SLC1, SLC6, SLC7, and SLC38 [24,25,26]. These glutamine transporters often share specificity with other neutral or cationic amino acids and have distinct transport modes. Some glutamine transporters are obligatory exchangers, while others function as active transporters in one direction. While most glutamine transporters import glutamine, some mediate the efflux of glutamine, which is coupled to the influx of other amino acids allowing for regulation of the intracellular pools of glutamine and amino acids. Once transported into the cell glutamine is used for the biosynthesis of glucosamines, nucleotides, and asparagine and may activate mammalian target of rapamycin complex 1 (mTORC1) in the cytoplasm. However, a majority of glutamine is metabolized to glutamate and ammonia (NH_3_) by the mitochondrial enzyme glutaminase (GLS) [27]. There are two isoforms of GLS, GLS1 and GLS2, but GLS1 is preferentially expressed by immune and vascular cells [28,29,30]. Interestingly, a recent report found that GLS1 also triggers mitochondrial fusion in a non-enzymatic manner, demonstrating that this enzyme regulates both mitochondrial metabolism and dynamics [31]. The GLS1 product NH_3_ induces the expression of heme oxygenase-1 (HO-1) and stimulates autophagy, which collectively allow cells to survive harmful stimuli, including inflammatory stress [32,33,34]. Furthermore, glutamate is exported from the mitochondria to the cytosol, where it is utilized to generate the tripeptide (glutamate, glycine, cysteine) antioxidant glutathione (GSH) and various non-essential amino acids such as alanine, aspartate, proline, ornithine, serine, cysteine, glycine, and arginine via the concerted action of several enzymes. While aspartate is utilized to generate nucleotides, arginine is metabolized to the critical signaling gas nitric oxide (NO) by NO synthase (NOS), see [22]. The release of NO by ECs plays a key role in maintaining vascular homeostasis. NO regulates blood flow and pressure by blocking arterial tone. It also exerts a potent antithrombotic effect by inhibiting platelet adhesion and aggregation. In addition, NO prevents neointima thickening and vascular occlusion by retarding SMC proliferation, migration, and extracellular matrix deposition. Moreover, NO suppresses inflammation by inhibiting the expression of adhesion receptors, the synthesis of proinflammatory cytokines and chemokines, and the recruitment, infiltration, and activation of leukocytes within blood vessels. Cytosolic glutamate can also be converted back to glutamine by glutamine synthetase (GS) at the expense of NH_3_- and adenosine triphosphate (ATP) [35]. Alternatively, mitochondrial glutamate is converted into alpha-ketoglutarate (αKG) by glutamate dehydrogenase 1 (GLUD1) or by the aminotransferase glutamic-pyruvic transaminase 2 (GPT2) and glutamic-oxaloacetic acid transaminase 2 (GOT2). Mitochondrial αKG participates in the tricarboxylic acid (TCA) cycle, supporting oxidative phosphorylation and the generation of electron donors (nicotinamide adenine dinucleotide (NADH) and flavin adenine dinucleotide (FADH_2_)) and ATP. However, under hypoxic conditions, αKG supports the reductive carboxylation pathway yielding citrate, which is used for fatty acid synthesis. In addition, αKG is transported to the cytosol, where it activates -mTORC1, promotes collagen synthesis via the activation of prolyl-4-hydroxlase, and serves as an important cofactor for nuclear enzymes involved in the epigenetic modification of histones and DNA. Thus, by replenishing TCA intermediates, glutamine works as an anaplerotic substrate, a privileged role that the amino acid plays in several types of normal and neoplastic cells.

## 3. Role of Glutamine in Immune Cells

Glutamine plays a major role in regulating the function of immune cells. The production of inflammatory cytokines by monocytes and macrophages is dependent on the availability of glutamine [36,37]. High concentrations of extracellular glutamine are required for maximal cytokine synthesis and phagocytosis by macrophages [38]. Furthermore, the optimal generation of NO by activated macrophages requires the presence of glutamine, which provides cells with additional substrate (arginine) for NO synthesis [39]. NO possesses potent microbiocidal properties, allowing for the clearance of viruses, bacteria, fungi, protozoa, and other foreign invaders [40]. Significantly, NO has been reported to mitigate the replication of SARS-CoV-2, possibly by nitrosylating and inhibiting the SARS-CoV-2 3CL cysteine protease [41]. In addition, glutamine influences the polarization of macrophages. Macrophages exhibit distinct phenotypic properties and are canonically classified into two functional subsets: M1 or classically activated and M2 or alternatively activated [42]. These two states represent extremes of a spectrum of macrophage activation states. While M1 macrophages respond to the initial phase of pathogen infection by generating cytotoxic quantities of NO, M2 macrophages are involved in the second phase of pathogen invasion and function to alleviate inflammation by redirecting the metabolism of arginine away from the synthesis of NO and towards the arginase-mediated production of polyamines and proline for tissue repair. Interestingly, glutamine is required for M2 macrophage differentiation [43]. In particular, the glutaminolysis-driven formation of αKG promotes M2 activation through metabolic and epigenetic reprogramming of M2 genes [44]. Thus, glutamine plays a crucial role in promoting M1 macrophage antiviral function by maximizing NO synthesis through the formation of arginine, while glutamine-derived αKG induces M2 macrophage activity to resolve inflammation and begin a healing response.

Glutamine also modulates the function of lymphocytes. Depletion of glutamine blocks the proliferation of activated T lymphocytes and their ability to release interleukin-2 and interferon-γ (IFNγ) [30,45]. Glutamine is also required for the growth, plasma cell differentiation, and antibody production of B lymphocytes [46]. In addition, glutamine is especially important for B cell survival in hypoxic conditions [47]. T cell activation induces a large increase in glutamine import and a rise in GLS1 expression. Notably, silencing GLS1 expression or pharmacological inhibition of GLS1 virtually abolishes T lymphocyte proliferation and cytokine production, highlighting the essential role of glutaminolysis in T cell activation [48,49]. Aside from regulating T lymphocyte activation, glutamine metabolism fosters the establishment of specific CD4 T cell subsets. The absence of glutamine or deficiency in glutamine uptake impairs both T helper 17 (Th17) and T helper 1 (Th1) T cell specification, and, instead, promotes the generation of regulatory T cells (Treg), which act to suppress the immune response [50]. Glutamine-dependent αKG deficiency converts Th1 cells to Treg cells [51] and deletes the gene GOT1, which metabolizes glutamate to αKG and converts Th17 cells to Treg-like cells by the epigenetic remodeling of the Foxp3 promoter region [52]. However, GLS1 deficiency selectively promotes Th1 and impedes Th17 differentiation, while not affecting Treg specification [53]. Subset-specific protein expression and different metabolites of glutamine may impart these discrete responses. Importantly, the altered profile of Treg cells in severe COVID-19 may arise from the low glutamine levels and the ensuing αKG deficiency [23]. Finally, glutamine deficiency or GLS1 inhibition inhibits clonal expansion and activation of CD8 T lymphocytes and may account for the exhaustion and depletion of these cells in patients with COVID-19 [54].

Glutamine may also influence the differentiation of myeloid-derived suppressor cells (MDSCs), which are elevated in COVID-19 [55]. However, experimental results are contradictory. While some studies revealed that low glutamine levels inhibit the differentiation of MDSCs, others found that glutamine deficiency stimulates the generation of MDSCs [56,57,58]. Nevertheless, glutamine may indirectly limit the production of MDSCs by blocking the synthesis of C-reactive protein, a potent inducer of MDSC formation [59,60]. In addition, glutamine exerts important effects on neutrophils. Glutamine protects neutrophils against apoptosis and impairs the chemotactic migration of neutrophils to sites of infection [61,62]. By blocking the secretion of interleukin-8, the amino acid may likewise prevent the release of neutrophil extracellular traps (NETosis), which combat SARS-CoV-2 infection but are linked to vessel occlusion and multiorgan damage in autopsied COVID-19 patients [63,64]. Finally, glutaminolysis is essential for the survival and growth of natural killer T (NKT) cells. GLS1 inhibition impairs NKT cell survival, proliferation, and activation, and this is associated with a decline in mitochondrial ATP production and GSH synthesis [65]. Thus, NKT cells adopt a glutamine-addicted phenotype to regulate their homeostasis and function.

## 4. Role of Glutamine in Vascular Health

Considerable evidence indicates that glutamine contributes to vascular health [21]. Cross-sectional clinical studies found that plasma glutamine or the glutamine/glutamate ratio is inversely related to blood pressure, hypertriglyceridemia, and insulin resistance, while plasma glutamate concentrations are linked to adverse vascular and metabolic indices [66,67,68]. In addition, a prospective study noted that the circulating glutamine/glutamate ratio is coupled to a reduced risk of cardiovascular disease, while blood glutamate is associated with an elevated risk, especially for stroke [69]. Interestingly, a genome association study discovered a genetic variant associated with diminished glutamine synthetase, which may result in a glutamine insufficiency, to the development of coronary artery disease in patients with type 2 diabetes [70]. Moreover, a large prospective study demonstrated that dietary intake of glutamine is inversely correlated to the risk of cardiovascular mortality, independent of other nutritional or lifestyle interventions [71]. Together, these studies illustrate the importance of glutamine in mitigating cardiovascular disease.

The mechanism(s) underlying the protective effect of glutamine in the circulation is not fully known. However, the ability of glutamine to directly influence the property of ECs is significant, as the dysfunction of these cells precedes the development of atherosclerosis and other cardiovascular disorders. Glutamine attenuates hyperglycemia-induced mitochondrial stress and apoptosis in human ECs [72] and preserves EC viability in response to oxidative stress, hypertonicity, and infection [73,74,75]. Glutamine also attenuates the expression of adhesion receptors on the surface of ECs and limits the migration of leukocytes across activated ECs [76,77,78]. In addition, glutaminolysis is required for energy production to drive ion transport in ECs, and the interruption of this pathway causes premature senescence of these cells [79,80]. Moreover, dietary supplementation of glutamine promotes the mobilization of endothelial progenitor cells in ischemic and diabetic rodents [81,82] and improves endothelium-dependent vasodilation in both mice and humans [83,84]. The latter findings are in-line with work showing that glutamine stimulates NO release from isolated arteries, possibly by providing blood vessels with additional arginine for NO synthesis [85]. Furthermore, the metabolism of glutamine protects ECs against the harmful effects of reactive oxygen species through the generation of GSH [86,87].

Recent studies also indicate that glutamine is indispensable for EC function [29,87,88,89]. The absence of glutamine or the inhibition of GLS1 expression or activity blocks the proliferation and migration of human ECs [87,88]. GLS1 inhibition arrests ECs in the G_0_/G_1_ phase of the cell cycle and this is associated with a pronounced decline in αKG and all TCA intermediates. However, exogenous supplementation of αKG replenishes TCA intermediates and partially rescues EC function, while the combination of TCA cycle replenishment and asparagine supplementation fully restores EC mitosis and movement. Thus, glutamine not only provides carbon to fuel the TCA cycle, but it also delivers nitrogen for asparagine synthesis to sustain cellular function. In addition, the selective loss of GLS1 in ECs inhibits the sprouting of vessels in various animal models of angiogenesis, illustrating the essential role of this enzyme in arterial expansion [86,87]. Significantly, intracellular glutamine concentrations determine the thrombogenicity of vascular cells [89]. Exogenous administration of glutamine elevates intracellular glutamine levels in vascular cells, and this results in diminished tissue factor expression and procoagulant activity, suggesting that glutamine within the blood vessel wall may be a promising target for controlling the thrombogenicity of inflamed or infected arteries. Consistent with this notion, glutamine prevents the activation of platelets, whereas glutamate switches platelets to an aggregatory phenotype through the activation of the alpha-amino-3-hydroxy-5-methyl-4-isoxazolepropionic acid (AMPA) receptor [90,91,92].

Interestingly, we discovered that glutamine-derived NH_3_ stimulates HO-1 gene transcription in ECs via the activation of nuclear factor erythroid 2-related factor 2 [32]. HO-1 is an important cytoprotective enzyme that metabolizes prooxidant heme into carbon monoxide (CO), iron, and biliverdin see [93,94,95,96]. While biliverdin and its downstream metabolite bilirubin are potent antioxidants, CO, much like NO, elicits many beneficial effects: it dilates blood vessels, inhibits platelet aggregation and vascular cell apoptosis, and exerts anti-inflammatory and anti-thrombotic actions in the circulation. In this regard, we recently reported that NH_3_ prevents EC death in response to inflammatory mediators via the HO-1-mediated release of CO [32]. This is consistent with a report showing that NH_3_ blocks the lethal action of tumor necrosis factor-α in renal epithelial cells, suggesting that this gas promotes cell survival in inflammatory states [33]. Furthermore, glutamine may limit tissue injury following ischemia-reperfusion in a HO-1-dependent manner, underscoring the importance of the NH_3_-HO-1-CO signaling axis in fostering vascular health [97].

## 5. Vascular Disease in COVID-19

COVID-19 is accompanied by a significant risk of ischemia-related vascular disease. Studies during the early phase of the pandemic found that the coordinated activation of inflammatory and thrombotic responses, thrombo-inflammation, is a primary cause of morbidity and mortality in patients with COVID-19 [98]. Indeed, laboratory studies detect the presence of a procoagulant state with elevated levels of circulating D-dimer and fibrinogen, a mild prolongation of prothrombin time in the plasma, and minimal thrombocytopenia in many patients hospitalized with COVID-19 [99,100,101,102]. Moreover, cross-sectional studies demonstrate that the rate of thromboembolic events such as thrombosis and pulmonary embolism are exceedingly high in critically ill patients with COVID-19 [103,104,105]. Post-mortem examinations of lung tissue in critically ill patients reveal a high frequency of platelet-fibrin thrombi in the small arteries and capillaries, suggesting a hypercoagulable condition that precipitates both venous and arterial thrombosis [105,106,107,108,109]. The elevated rate of arterial thrombosis in COVID-19 possibly plays a role in the higher frequency of myocardial infarction, ischemic stroke, and acute limb ischemia in this patient cohort [104,110,111,112]. Autopsy findings also indicate a high prevalence of alveolar microthrombi in patients with COVID-19. These occluding microthrombi extend beyond the lungs to the heart, kidney, and liver, suggesting the widespread presence of thrombotic microangiopathy in COVID-19 [113]. More recently, a global, multicenter database analysis confirmed the occurrence of disseminated intravascular coagulopathy and thrombosis among critically ill patients with COVID-19 [114].

The etiology of thrombosis in patients with COVID-19 is multifactorial involving complement activation and cytokine release, coagulation abnormalities, platelet hyperactivity and apoptosis, the formation of neutrophil extracellular traps, and endothelial dysfunction [5,113]. However, endothelial malfunction is a central feature in COVID-19 thrombotic complications. In particular, the endothelium undergoes a prothrombotic transformation involving the loss of the glycocalyx and cytoprotective signaling, and the generation of thrombotic effectors to promote fibrin formation, platelet adhesion, and complement activation [115]. Clinical signs of endothelial inflammation and death in COVID-19 are common and found in multiple organs [98,116,117]. Autopsy studies on lung specimens from SARS-CoV-2-infected patients detected severe EC damage with evidence of apoptosis and loss of tight junctions [118]. In addition, biomarkers of endothelial dysfunction are elevated in patients with COVID-19. Plasma von Willebrand factor (vWF) is increased in COVID-19 patients with high concentrations associated with severe disease [119,120,121]. Moreover, circulating levels of P-selectin, E-selectin, soluble intercellular adhesion molecule-1 (ICAM-1), thrombomodulin, angiopoietin-2, and plasminogen activator inhibitor-1 (PAI-1) are augmented in COVID-19 [98,122,123]. The release of vWF, selectins, and ICAM-1 following EC activation binds platelets, neutrophils, and monocytes to initiate thrombosis, while the shedding of thrombomodulin by pro-inflammatory cytokines would further promote the procoagulant and inflammatory milieu within the vasculature of COVID-19 patients [98]. In addition, the release of PAI-1 by ECs would reduce fibrinolysis by inhibiting the tissue plasminogen activator, thereby stimulating fibrin deposition in the microvasculature. Circulating ECs, which reflect cells that are detached from the damaged blood vessel, are also higher in critically ill patients and may contribute to the enhanced vascular permeability in patients with COVID-19 [124]. A recent meta-analysis disclosed that several biomarkers of endothelial dysfunction are significantly associated with increased composite poor outcomes in COVID-19 patients [125]. Collectively, these observations attest to the severe endotheliopathy and coagulopathy that is present in COVID-19. Although preliminary work implicated the virus in promoting EC dysfunction, recent studies suggest an indirect mechanism mediated locally and systemically by excessive cytokine synthesis [116].

COVID-19 also negatively impacts vascular tone. Endothelium-dependent microvascular reactivity in the skin is severely restricted in critically ill COVID-19 patients [126]. In addition, systemic endothelium-dependent vasodilator responses are markedly decreased in patients with severe or mild to moderate COVID-19 relative to healthy individuals, and this is paralleled by an increase in circulating inflammatory cytokines and chemokines [127]. Moreover, endothelium-dependent vasodilation remains impaired shortly after acute COVID-19 but improves following long-term recovery from the disease [128,129]. Together, these studies suggest that NO bioavailability is compromised in COVID-19 patients. This is consistent with reports showing a decrease in serum NO metabolites in COVID-19 patients, most likely due to the consumption of NO by reactive oxygen species [130,131]. In addition, the degradation of the endothelial glycocalyx in COVID-19 may also contribute to the decline in NO synthesis since it is critically involved in flow-mediated NO production [132,133,134]. The reduction in endothelial NO may further aggravate ischemic injury in COVID-19 by promoting blood vessel spasm and eliminating a key brake on platelet activation and aggregation. Moreover, the loss of NO-mediated vasodilation is further amplified by an angiotensin converting enzyme (ACE)/ACE2 imbalance in COVID-19 that favors angiotensin II-mediated vasoconstriction [135]. In fact, spontaneous and severe coronary vasospasm has been reported in patients with COVID-19 [136,137,138].

## 6. Glutamine Deficiency in COVID-19

Metabolomic studies have consistently identified a decline in circulating levels of glutamine in multiple patient populations with COVID-19 that is correlated with disease severity [23,139,140,141,142,143,144,145,146,147]. COVID-19 patients have a significantly reduced glutamine to glutamate ratio, indicating the increased utilization of glutamine. Consistent with this notion, reductions in glutamine are accompanied by an increase in plasma GLS activity [148,149,150]. Furthermore, the decrease in glutamine concentration in severe COVID-19 patients is positively correlated with lactate dehydrogenase, C-reactive protein, and pCO_2_ and positively correlated with pO_2_, disclosing formerly unknown consequences of low glutamine in severe COVID-19 pathologies [151]. Moreover, meta-analysis showed that elevated glutamine is related to a decreased risk of COVID-19 infection and severe COVID-19, whereas raised glutamate levels are associated with increased risk of infection and serious disease [152]. Intriguingly, an advanced bioinformatic platform reported that glutamine was the top candidate of over 26,000 FDA-approved drugs tested for reversing coronavirus associated alterations in gene expression [153]. In total, these studies underscore the potential therapeutic importance of restoring glutamine levels in COVID-19 patients.

## 7. Strategies Targeting Glutamine in COVID-19

There is an emerging realization that amino acids play an essential role in preserving immune function and vascular health [21,22,154,155,156,157,158]. Considerable evidence indicates that the bioavailability of glutamine is critically depleted in COVID-19 patients, perhaps secondary to a rise in GLS activity that directs the conversion of glutamine to glutamate (Figure 2). The resulting decline in glutamine will compromise EC function and NO production by limiting cellular arginine levels. The loss of glutamine also stimulates vascular inflammation, vascular cell thrombogenicity, platelet activation and aggregation, and reduces the viability of ECs by restricting NO synthesis and the induction of HO-1. In addition, glutamine deficiency causes a broad dysfunction of the immune system that worsens the gravity of COVID-19. The consumption of glutamine in COVID-19 will also increase oxidative stress by reducing the synthesis of GSH and the expression of HO-1. Collectively, these actions related to glutamine insufficiency will promote vasospasm, thrombosis, and NETosis, resulting in vascular occlusion and organ failure. Crucially, comorbidity-associated glutamine deficiency is a predisposition to severe COVID-19 [159].

Multiple approaches may be used to mitigate the loss of glutamine in COVID-19. Dietary supplementation of glutamine has proven effective in correcting deficiencies in circulating glutamine in cardiometabolic disease and hemolytic disorders and is now prescribed in patients with sickle cell disease [160,161,162]. However, glutamine generates ATP and precursors for the synthesis of macromolecules required for virus assembly, and restoration of this amino acid may augment SARS-CoV-2 replication and infection. In fact, host cell glutamine metabolism has been identified as a potential treatment against COVID-19 [163,164]. Nevertheless, preliminary clinical studies support the use of glutamine in COVID-19 patients. An initial small, retrospective, cross-sectional study found that the oral ingestion of glutamine powder (30 g/day) with meals shortens hospital stays and lessens the need for intensive care in patients with COVID-19 [165]. In addition, a single-blind randomized clinical trial reported that intravenous glutamine administration (0.4 g/kg/day) lowers the inflammatory response in COVID-19 patients admitted to the intensive care unit but does not improve short-term mortality in this patient cohort [166]. More recently, a larger case–control study noted that the short-term meal-time consumption of glutamine (30 g/day) reduces serum markers of inflammatory and oxidative stress and increases appetite in hospitalized COVID-19 patients [167]. A major limitation of current studies is that it is not known whether glutamine administration elevates glutamine availability in COVID-19 patients. Due to the extensive metabolism of glutamine by the splanchnic circulation, high doses of glutamine may be needed to fully restore plasma glutamine in these patients [168]. While glutamine is usually administered using the free form, the use of more stable dipeptide formulations, including L-glycyl-L-glutamine, L-arginyl-L-glutamine, and L-alanyl-L-glutamine, should be considered since they possess a more favorable pharmacokinetic profile [168,169]. The use of L-arginyl-L-glutamine in COVID-19 is especially appealing since deficits in both glutamine and arginine have been detected in this disease [23,170,171]. While a plethora of studies have confirmed the safety of dietary glutamine supplementation, care should be implemented when using this amino acid in critically ill patients [172,173]. Clearly, larger, multi-center dose-escalation studies employing various preparations of glutamine are needed to clarify the safety, utility, and dosing requirements for this amino acid in patients with COVID-19. The prophylactic use of glutamine in individuals with a high risk for poor outcomes following SARS-CoV-2 infection should also be considered.

The administration of specific metabolites of glutamine with known beneficial actions should also be considered. In this respect, the thoughtful delivery of NH_3_ may be advantageous given its ability preserve to EC viability and stimulate HO-1 expression. In addition, NH_3_ and its chloride salt inhibits the replication of several viruses, including the influenza virus, reovirus, papilloma virus, and the infectious pancreatic necrosis virus [174,175,176,177], and may be efficacious against the SARS-CoV-2 virus [178]. Aside from NH_3_, the use of another gas such as NO holds promise in managing COVID-19 [179,180]. Both inhaled NO and NO donors have a wide range of antiviral activity and serve as a first line of defense against foreign invaders. In addition, NO plays a critical role in maintaining vascular integrity and endothelial function, controlling the response of immune cells and platelets, and has pulmonary protective properties. Furthermore, the glutamine metabolite αKG may be of benefit to patients with COVID-19. αKG has immuno-augmenting properties and a recent report found that ingestion of αKG inhibits SARS-CoV-2 replication and reduces inflammation, thrombosis, and apoptotic cell death in the lungs of in infected animals [181]. Moreover, αKG restores EC function in cells deprived of glutamine, and oral administration of αKG enhances NO synthesis by ECs in diet-induced obese rodents [26,84,85,182]. Thus, the delivery of NH_3_, NO, and αKG provides an attractive means for combatting COVID-19.

Alternatively, downstream effectors of glutamine may be employed to treat COVID-19. In this respect, HO-1 represents an attractive target for COVID-19 since it is strongly induced by glutamine and possesses potent antiviral and vasoprotective properties [32,183,184,185]. HO-1 or its reaction products CO and biliverdin/bilirubin suppress many viral infections, including influenza A, hepatitis B, human immunodeficiency virus, Ebola, dengue, and Zika, and there is a strong possibility that they may control SARS-CoV-2 infections by restoring IFNγ production and/or blocking viral proteases and RNA polymerases [186,187,188]. In addition, HO-1 and its end products confer anti-thrombotic effects by lessening endothelial injury, the expressing of adhesion receptors, and inflammatory responses, diminishing procoagulant molecules, such as vWF, PAI-1, and tissue factors [93,94,95,96,97,189,190,191,192,193]. Interestingly, an increase in HO-1 expression is observed in critically ill COVID-19 patients, and this may serve as an adaptive mechanism to counteract increased heme levels driving coagulation and thrombosis in these patients [194,195].

Several approaches may be taken to target HO-1 in COVID-19. Numerous inducers of HO-1 have been identified and shown to be effective in preclinical studies. Heme and its derivates are potent inducers of HO-1 that elicit beneficial effects in animal models of infection, inflammation, and cardiovascular disease [93,94,95,96,97,186]. However, the induction of HO-1 by heme analogues in patients is transient and does not persist for more than one week [196]. Moreover, heme is a danger-associated molecular pattern molecule that may exacerbate inflammation in SARS-CoV-2-infected patients [197]. More promising, the synthetic triterpenoid bardoxolone methyl is a strong inducer of HO-1 that has generated some positive results in patients with kidney disease [198,199,200]. In addition, dimethylfumarate is another activator of HO-1 that is approved for use in multiple sclerosis and, importantly, shows efficacy in an animal model of vascular occlusion [201]. Interestingly, we recently identified the sodium–glucose cotransporter 2 inhibitor canagliflozin as a robust inducer of HO-1 that directly mitigates vascular cell proliferation and inflammation [190,202]. This work suggests that HO-1 contributes to the beneficial cardiovascular profile of canagliflozin in patients with type 2 diabetes and supports the use of this drug in both the diabetic and non-diabetic patient populations [203].

The direct application of HO-1-derived products affords another avenue for ameliorating COVID-19-mediated complications. While the inhalation of CO has been successfully employed in a myriad of preclinical studies [93,94,95,96,204], its translation to the clinic remains problematic, owing to safety concerns and suboptimal dosing regimens [205]. The difficulties associated with CO inhalation protocols led to the development and use of CO-releasing molecules that liberate controlled amounts of CO in response in specific stimuli [206,207]. In addition, the use of organic-based click-and-release prodrugs that employ a chemical reaction to generate CO provides another mode for the application of this gas [208]. Furthermore, saturated solutions of CO impart a simple vehicle for gas delivery that has been demonstrated to prevent vascular occlusion in injured rat arteries and in a murine model of sickle cell disease [209,210]. In a similar manner, saturated solutions of biliverdin may also be used to attenuate COVID-19-related issues [211,212,213]. However, the limited solubility and stability of bilirubin in aqueous solutions hampers its direct administration. Instead, a pegylated form of bilirubin that self-assembles into hydrophilic nanoparticles may be used, as this formulation was shown to be non-toxic and effective in a mouse model of ulcerative colitis [214]. One potential drawback with the use of bile pigments is their rapid metabolism by the liver which necessitates frequent dosing. However, the rise in circulating bilirubin following its administration may be extended by blocking its metabolism by uridine diphosphate-glucuronosyltransferase 1A1 (UGTA1A). Indeed, atazanavir and canagliflozin are dual HO-1 inducers and UGTA1A inhibitors that raise plasma bilirubin concentrations and improve vascular function in diabetic patients, offering proof of principle for this strategy [191,202,215,216,217,218].

## 8. Conclusions

Clinical studies have discovered a low circulating level of glutamine in patients with COVID-19 that is associated with disease severity. This glutamine deficiency is a central metabolic feature of COVID-19 that may contribute to the increased risk of infection, inflammation, immune and EC dysfunction, coagulopathy, vascular occlusion, and multi-organ failure. Strategies that target glutamine, its metabolites, and/or its downstream effectors along with approved antiviral drugs may provide a robust approach in alleviating the calamitous ramifications of SARS-CoV-2 infection.

## Figures and Tables

**Figure 1 ijms-24-07593-f001:**
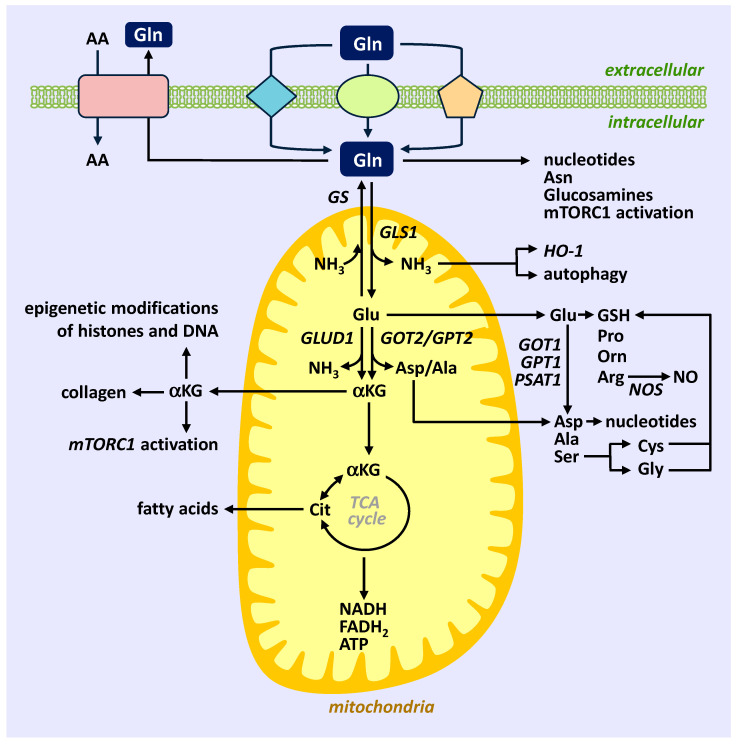
Summary of Gln transport and metabolism in immune and vascular cells. Gln is transported into cells through several membrane transporters and modulates the entry and intracellular concentration of other AA through its interaction with Gln transporters. Following uptake, Gln is utilized in the cytosol to generate nucleotides, Asn, and glucosamines and may activate mTORC1. A majority of glutamine is converted to Glu and NH_3_ by GLS1 in the mitochondria. NH_3_ promotes cell survival by stimulating autophagy and the induction of HO-1. Glu that is exported to the cytosol is used to generate GSH and numerous non-essential amino acids via the action of a plethora of enzymes. The formed Arg is metabolized to NO by NOS, while Asp is used for nucleotide synthesis. Cytosolic Glu can also be converted back to Gln by GS. Mitochondrial glutamate is deamidated by GLUD1 or various aminotransferases to αKG, which supplies metabolites to the TCA cycle to fuel the generation of NADH, FADH_2_, and ATP. Under hypoxic conditions, αKG supports the reductive carboxylation pathway yielding citrate, which is used for fatty acid synthesis. Gln-derived αKG also stimulates mTORC1 activity and collagen synthesis and influences the epigenetic modification of histones and DNA. The different colored shapes (ellipse, rhombus, pentagon, and quadrilateral) represent distinct Gln transporters. AA, amino acids; Gln, glutamine; Asn, asparagine; Glu, glutamate; GS, glutamine synthetase; GLS1, glutaminase-1; NH_3_, ammonia; HO-1, heme oxygenase-1; GSH, glutathione; Pro, proline-; Orn, ornithine; Arg, arginine; Asp, aspartate; Ala, alanine; Ser, serine; Cys, cysteine; Gly, glycine; NO, nitric oxide; NOS, nitric oxide synthase; GLUD1, glutamate dehydrogenase 1; GOT1/2, glutamic-oxaloacetic acid transaminase 1/2; GPT1/2, glutamic-pyruvic transaminase 1/2; PSAT1, phosphoserine aminotransferase 1; αKG, alpha-ketoglutarate; TCA, tricarboxylic acid; NADH, nicotinamide adenine dinucleotide; FADH_2_, flavin adenine dinucleotide; ATP, adenosine triphosphate; mTORC1, mammalian target of rapamycin complex 1.

**Figure 2 ijms-24-07593-f002:**
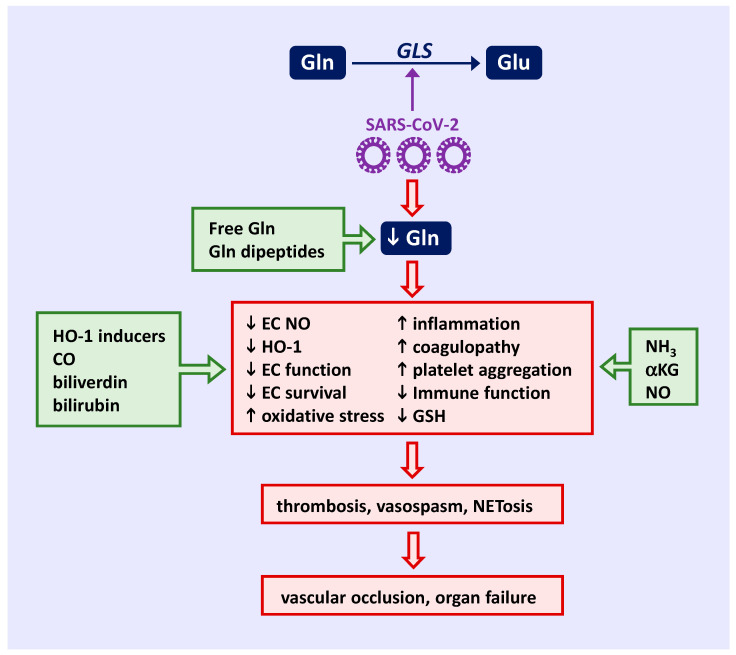
Targeting Gln in COVID-19-mediated immune and endothelial dysfunction. SARS-CoV-2 lowers circulating levels of Gln by increasing GLS activity. The resulting decline in Gln compromises EC function and NO production by limiting cellular Arg levels. The loss of Gln also stimulates vascular inflammation, platelet aggregation, coagulopathy, and reduces the viability of ECs by restricting NO synthesis and the induction of HO-1. In addition, Gln deficiency causes a broad dysfunction of the immune system that worsens the severity of infection. The consumption of Gln in COVID-19 will also increase oxidative stress by reducing the synthesis of GSH and expression of HO-1. Collectively, these actions will promote vasospasm, thrombosis, and NETosis, resulting in vascular occlusion and multi-organ failure. Several strategies may be employed to target glutamine in COVID-19. Dietary supplementation with free or dipeptide forms of Gln provides a straightforward approach to restore circulating levels of Gln in SARS-CoV-2-infected patients. Alternatively, the direct administration of Gln metabolites (NH_3_ or αKG) or downstream effectors of Gln (inhaled NO, NO donors, HO-1 inducers, CO donors, biliverdin, or bilirubin) provides a more selective avenue in correcting disturbances of Gln metabolism in patients with COVID-19. Downward pointing arrows indicate a decrease while upward pointing arrows indicate an increase. Gln, glutamine; Glu, glutamate; COVID-19, coronavirus disease 2019; SARS-CoV-2, severe acute respiratory syndrome coronavirus 2; GLS, glutaminase; EC, endothelial cell; NO, nitric oxide; Arg, arginine; HO-1, heme oxygenase-1; GSH, glutathione; NETosis; the release of neutrophil extracellular traps; NH_3_, ammonia; αKG, alpha-ketoglutarate; CO, carbon monoxide.

## Data Availability

Not applicable.

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
