# Peer review of "Glutamine Deficiency Promotes Immune and Endothelial Cell Dysfunction in COVID-19"

_ijms, 2023, doi:10.3390/ijms24087593_

Round 1
Reviewer 1 Report
The review by William Durante is interesting and presents a complete recount of the experimental data available. Some remarks:
1) Gln metabolism, Fig. 1. In the legend, it is correctly stated that several transporters provide transmembrane transport of Gln. Yet, in the figure only a single transporter is shown, giving the (unexperienced) reader an oversimplified picture of Gln transport. Actually, Gln transport is an important regulatory step in Gln metabolism, in particular under stress or pathological conditions, when several of these transporters show changes in expression/activity. I suggest (1) to slightly modify the Figure, adding some other, differently shaped transporters to sketch the complexity of Gln transport and (2) to integrate the text with some concise information on Gln transporters. This could be really important, since, through its interaction with multiple transporters, Gln can actually modulate the entry and the intracellular concentration of other amino acids, a function that should be added to its many roles. Moreover, due to its interaction with secondary active transporters, it can work as a cytoprotective compatible osmolyte regulating cell volume.
2) The discussion at p. 3 and 4 about the role of Gln in cell metabolism is fairly complete. I have two suggestions. First, the description of Gln in fueling the Krebs cycle could be integrated stating that, for these reasons, Gln can work as an anaplerotic substrate, a privileged role that the amino acid plays in several types of both normal and neoplastic cells. Second, beside aKG, experimental evidence suggests that also Gln itself may activate mTORC1.
3) I am confused by the discussion reported at lines 419-422. If you block/inhibit GLS to decrease the conversion of Gln to Glu, how can you expect to promote GSH synthesis? Usually, as also reported in the review, Gln-dependent promotion of GSH synthesis is attributed to Gln-derived Glu. Please justify or modify the sentence.
Minor
Language is accurate and I have found very few typos.
Terminology should be homogenized. For instance, the virus is named SARs-CoV-2 at the beginning of the text and SARS-CoV-2 in the rest of the review. In PO2 and PCO2, “2” should be subscript. Most commonly, CRP is indicate as “C-reactive protein” and not as “c-reactive protein”.
At line 354, the second “glutamine” is obviously “glutamate”.
Author Response
I would like to thank the reviewer for their careful analysis of the manuscript. I am pleased that they found the review interesting and that it presents a complete recount of the experimental data. In addition, the reviewer raised a number of important issues that we address below.
- I agree with the reviewer that Gln transport is an important step in Gln metabolism. As suggested, I have slightly modified Figure 1 to show that Gln transport occurs via multiple transporters (added differently shaped transporters) and integrated the text with some concise information on Gln transporters. In addition, I indicate that Gln can modulate the entry and intracellular concentrations of other amino acids via its interaction with Gln transporters.
- As suggested, I've integrated the description of Gln fueling the TCA cycle by stating that Gln works as anaplerotic substrate, a privileged role that the amino acid plays in several types of normal and neoplastic cells at the end of the discussion describing the role of Gln in cell metabolism. In addition, I now mention and show in Figure 1 that Gln itself may activate mTORC1.
- I agree with the reviewer and have deleted the discussion using GLS1 inhibitors as it is likely to compromise GSH synthesis.
- I apologize for the inconsistent use of terms and the typos. I have now homogenized the term SARS-CoV-2 throughout the manuscript and corrected pO2, pCO2, and C-reactive protein.
Reviewer 2 Report
This is a nice and comprehensive review on the impact of glutamine on immune and endothelial (dys)funtion with a focus on COVID-19.
The review is very well written, provides a lot of simply explained information on the biology of glutamine and related molecuels such as NH3, CO, NO, HO-1.
There is however one reference lane 256, which in my oppinion is not appropriate. The paper referenced with 91 is a review and the sentence in the lanes 255-256 would suggest the information is derived from a reserch article but not a review paper.
Please control this and other references for compatibility with the corresponding text.
Author Response
I would like to thank the reviewer for their thorough review of our manuscript. I am pleased that they found the review to be well written and comprehensive.
The reviewer correctly noted that reference 91 is a review and not derived from a research article as suggested by the corresponding text. We apologize for this error as I incorrectly cited the wrong reference. In the revised manuscript, we now correctly cite the appropriate research article (new ref#32). We have also checked other references for compatibility with the corresponding text.